# TAaCGH Suite for Detecting Cancer—Specific Copy Number Changes Using Topological Signatures

**DOI:** 10.3390/e24070896

**Published:** 2022-06-29

**Authors:** Jai Aslam, Sergio Ardanza-Trevijano, Jingwei Xiong, Javier Arsuaga, Radmila Sazdanovic

**Affiliations:** 1Department of Mathematics, NC State University, Raleigh, NC 27695, USA; jkaslam@ncsu.edu; 2Department of Physics and Applied Mathematics, University of Navarra, 31008 Pamplona, Spain; sardanza@unav.es; 3Institute for Data Science and Artificial Intelligence, University of Navarra, 31009 Pamplona, Spain; 4Graduate Group in Biostatistics University of California Davis, Davis, CA 95616, USA; jwxxiong@ucdavis.edu; 5Department of Molecular and Cellular Biology, University of California Davis, Davis, CA 95616, USA; 6Department of Mathematics, University of California Davis, Davis, CA 95616, USA

**Keywords:** breast cancer molecular subtypes, genomics, topological data analysis, CNA copy number aberrations, lifespan curves, Betti curves, persistence landscapes

## Abstract

Copy number changes play an important role in the development of cancer and are commonly associated with changes in gene expression. Persistence curves, such as Betti curves, have been used to detect copy number changes; however, it is known these curves are unstable with respect to small perturbations in the data. We address the stability of lifespan and Betti curves by providing bounds on the distance between persistence curves of Vietoris–Rips filtrations built on data and slightly perturbed data in terms of the bottleneck distance. Next, we perform simulations to compare the predictive ability of Betti curves, lifespan curves (conditionally stable) and stable persistent landscapes to detect copy number aberrations. We use these methods to identify significant chromosome regions associated with the four major molecular subtypes of breast cancer: Luminal A, Luminal B, Basal and HER2 positive. Identified segments are then used as predictor variables to build machine learning models which classify patients as one of the four subtypes. We find that no single persistence curve outperforms the others and instead suggest a complementary approach using a suite of persistence curves. In this study, we identified new cytobands associated with three of the subtypes: 1q21.1-q25.2, 2p23.2-p16.3, 23q26.2-q28 with the Basal subtype, 8p22-p11.1 with Luminal B and 2q12.1-q21.1 and 5p14.3-p12 with Luminal A. These segments are validated by the TCGA BRCA cohort dataset except for those found for Luminal A.

## 1. Introduction

Cancer is a set of polygenic diseases that are partly driven by chromosome aberrations in the form of copy number [1,2,3]. Copy number changes are found in most cancers and since they are known to be key regulators of gene expression, are frequently used as markers to identify cancer-driving genes [4]. Microarray and sequencing technologies have been used to detect copy number changes in cancer [5,6,7,8]. These experimental methods have identified key cancer driver genes and revealed that not all copy number changes drive gene expression. Instead, many copy number changes that are apparently unrelated to gene regulation also accumulate as cancer evolves. The accumulation of these “passenger” copy number aberrations make the identification of cancer-driving copy number changes a challenging task.

The standard approach to differentiate cancer-driving copy number aberrations from passenger copy number changes is through association studies. Measurements from topological data analysis have been used effectively as inputs into statistical methods to detect copy number changes [9]. Topological data analysis has more generally been successful in cancer genomics. For example, the Mapper algorithm was used to detect a new subgroup of breast cancer with a high survival rate [10]. More recently, it has been used to detect potential tumor-producing genes some of which have been confirmed in mouse models [11].

In a typical study, array Comparative Genomic Hybridization data (aCGH) [12], is first segmented (most commonly using circular binary segmentation or similar approaches [13]) and each of the segments is then tested for association to a previously selected phenotype. In [9], a topological data analysis method was introduced to identify copy number changes associated with a given cancer phenotype. In this approach, called Topological Analysis of array CGH (TAaCGH), copy number data measured using array CGH is mapped into a point cloud from which topological signatures are extracted. Then an association study between these topological signatures and the desired phenotype is conducted. Figure 1 shows the TAaCGH workflow. One difference between topological approaches and traditional approaches is that their multiscale character allows for multiresolution analysis of each copy number change, making pre-selected cutoffs unnecessary. Additionally, because of the global character of topological signatures, they can capture combined effects from different copy number changes, as is expected in a polygenic disease.

TAaCGH was used to identify copy number changes associated with breast cancer molecular subtypes [14]. The study used Betti curves in dimension 0, $\beta_{0}$ curves, to detect copy number changes and it showed an overall agreement with other current methods to identify chromosome aberrations. This study was later combined with statistical learning methods to build logistic regression models as classifiers for cancer subtypes [15]. Betti curves in dimension 1 were also studied in [16] and used to identify co-occurring copy number changes, that is copy number changes that tend to appear in combination with other copy number changes but do not appear independently as observed in [17,18].

Betti curves are known to be unstable with respect to small perturbations of the data [19]. We provide bounds on the distance between the Betti and lifespan curves that come from the Vietoris–Rips complex and the one arising from the time series with a slight perturbation for both Betti and lifespan curves (Section 3) leveraging results from [20,21].

Given copy number data, which we treat as time series data with position along the genome taking the place of time, we build a Vietoris–Rips complex on the associated sliding window point cloud. We expand the TAaCGH method to lifespan and persistent landscape curves and perform an exhaustive comparison of the performance of these curves on simulated and patient data [22]. Lifespan and persistent landscape curves have the potential to capture different properties of data than Betti curves. This was shown in [23], where the authors compared the persistent entropy function, Betti curves and other related summaries. They showed that these curves provide complementary information for the task of image classification. We therefore hypothesize that other persistence curves may discover relevant genes complementary to the ones found by Betti curves in [14].

Simulation results (Section 4.1) indicate that lifespan curves outperform Betti and landscape curves for the task of distinguishing a group of patients with single contiguous aberrations from a group of patients with no aberrations. In particular, lifespan curves are less sensitive to noise in the data of individual patients than the other persistence curves. Additionally, lifespan curves perform better on focal aberrations than Betti or landscape curves.

The performance of TAaCGH with lifespan and landscape curves on the dataset published by Horlings and colleagues [22] is presented in Section 4.2. The performance of all curves is comparable in detecting significant regions across molecular subtypes. In the HER2 subtype, all three curves detected segments in 17q, the long arm of chromosome 17, and importantly they all detected cytobands 17q12-q21.31 which contain the ERBB2 gene. For the Luminal A subtype, Betti and lifespan curves detected a subset of the regions detected by persistence landscapes. In the Luminal B subtype only one region of the short arm of chromosome 8, 8p22-p11.1, was detected and it was detected by Betti curves. In the Basal subtype the three curves detected similar regions, with landscapes detecting some different regions from the other two curves. Newly detected regions include 1q21.1-q25.2, 2p23.2-p16.3, 23q26.2-q28 for the Basal subtype, 8p22-p11.1 for Luminal B and 2q12.1-q21.1 and 5p14.3-p12 for Luminal A. The TCGA BRCA cohort data supports the new regions associated with the Basal and Luminal B phenotypes.

As in [15], we build predictive models using logistic regression and find that all approaches perform similarly, despite finding different predictor variables. In the Luminal A subtype only the region 5p14.3-p12 was found to have predictive power. For HER2 either one of overlapping segments 17q11.1-q12 and 17q12-q21.31 was found to have predictive power depending on the persistence curve used. The predictor variables for Basal predictive models differed greatly between persistence curves. The only repeated predictor variable was 10p12.31-p11.1 which was a predictor variable for the lifespan and fourth landscape curve predictive models.

**Figure 2 entropy-24-00896-f002:**
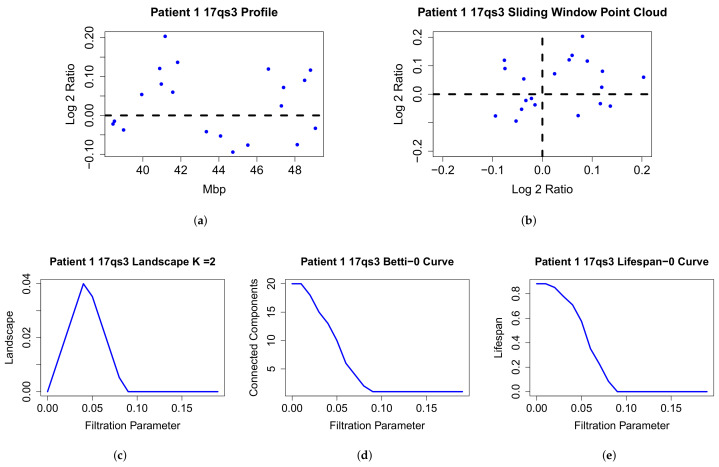
Sample patient data and output. The copy number data (**a**), sliding window point cloud (**b**), persistent landscape (**c**), Betti curve (**d**) and lifespan curve (**e**) of a patient from the Horlings dataset [22] on chromosome 17q segment 3 using 0-dimensional persistence. 17qs corresponds to the cytoband range 17q21.2-q21.33.

## 2. Data and Methods

### 2.1. Persistence Curves

In this section, we discuss persistence curves, an important tool for summarizing topological information, which we then combine with statistical methods.

Since the number of points in persistence diagrams varies based on the values of the data there is no well-defined mean of persistence diagrams and they cannot be treated as a fixed-length vector. This makes methods from statistics and machine learning difficult to apply. In order to overcome this, the topological information from persistent homology is frequently summarized using tools such as persistence curves [20], kernel SVM for persistence [24], persistence landscapes [25], and persistence images [26] among others. In this paper, we focus on persistence curves including Betti and lifespan curves, and persistence landscapes. It is worth noting that, in the 0-dimensional case, one generator could have an infinite lifespan. We therefore choose to consider reduced persistent homology for lifespan curves which amounts to removing the infinite generator in dimension 0 or assert that the infinite generator dies at a predetermined filtration value.

Given an *n*-dimensional persistence diagram *D* the *n*th Betti curve, denoted $\beta_{n}\left(D,t\right)$, is equal to the number of birth-death pairs (b,d)∈D such that t∈(b,d]. Similarly, the *n*th lifespan curve denoted $\ell_{n}\left(D,t\right)$ is equal to the sum of the lifespans of all birth-death pairs (b,d)∈D such that t∈(b,d]. Persistent landscapes are a form of persistence curve introduced in [25]. The *k*th persistence landscape of *D*, denoted λ(k,t), is $\lambda \left(k,t\right)={\mathrm{kmax}}_{p}\left({\left[\min\left(t-b,d-t\right)\right]}_{+}\right)$ where p=(b,d)∈D, ${\left[c\right]}_{+}=\max\left(c,0\right)$ and kmax is the *k*th highest value.

Persistence curves were introduced as a general framework under which previously studied summaries of persistence diagrams lie [20]. A similar framework was also built and studied in [21]. The work in [20] allows for easy generation of new summaries including lifespan curves, which we consider. Lastly, the persistence curve framework provides a way to make general arguments about the stability of the curves with respect to the bottleneck distance between persistence diagrams. Persistence landscapes were shown to be stable in [25].

### 2.2. The Topological Analysis for Array CGH (TAaCGH) Method

The TAaCGH method is a form of genetic association study which uses copy number data to associate segments of the genome with particular subtypes of breast cancer. It differs from standard association studies by using topological information as a test statistic. An overview of the pipeline is pictured in Figure 1. The method begins by splitting the copy number data from each chromosome arm into consecutive segments of 20 probes each. Each segment overlaps with the next segment in 10 probes. Next, the data set is split into a test and control set. The process tests for the significance of each individual segment. The test set consists of all patients from one cancer subtype and the control set consists of all other patients. The data for a fixed segment is converted into a point cloud in $R^{2}$ using the sliding window mapping [9,27] for each patient. A Vietoris–Rips complex is built on each point cloud and the persistent homology of each Vietoris–Rips complex is taken. The persistent homology is then summarized into persistence curves for each patient. The data, sliding window point cloud, Betti curve, lifespan curve, and 2nd persistent landscape curve on 0-dimensional persistence are pictured in Figure 2 for a particular patient. Next, the persistence curves from all patients in the test set are averaged together and similarly for the control set. The $L^{2}$ norm of the difference between the average test and the control persistence curves is used as the test statistic. Permutation testing with FDR correction for multiple testing is used to test the significance of the statistic and thus of each segment for each cancer subtype.

It is worth noting that in [14], persistence was calculated using JavaPlex [28] which in some cases outputs a shorter lifespan than the R TDA package used in this study. To illustrate the difference, consider 1-dimensional persistence of a Vietoris–Rips filtration on the four vertices of a unit square. If the JavaPlex discretization increment is set to 0.01, a 1-dimensional cycle connecting all the vertices is born at filtration parameter value t=1 which agrees with the R TDA output. However, R TDA computes its persistence as precisely $\left[1,\sqrt{2}\right)$ while JavaPlex gives [1,1.41), due to the choice of discretization.

### 2.3. Horlings Dataset

The dataset used in this study is from [22]. This is the same dataset used in [9,14,15]. It consists of BAC Microarrays from the genome with an average spacing of 1 Mb. Each BAC clone was spotted in triplicate on each slide (Code Link Activated Slides, Amersham Biosciences). The dataset contains 68 patient samples from the 4 most common molecular subtypes of breast cancer: Luminal A, Luminal B, basal-like and HER2+. There are 21 Luminal A samples, 12 Luminal B samples, 21 basal-like samples and 14 HER2+ samples. We consider each molecular subtype separately as the test group and the remaining patients as the control group.

### 2.4. TCGA BRCA Cohort Data

The TCGA BRCA cohort data was collected from the Firehose dataset with the disease name: breast invasive carcinoma [29]. The dataset consists of 1098 tumors hybridized to the Affymetrix SNP 6.0 array platform, using the GRCh38 assembly of the human genome as a reference. Circular binary segmentation (CBS) was applied and the copy number values were estimated for each segment. Results were then $log_{2}\left(copynumber/2\right)$ transformed and used to assign focal scores to protein-coding genes. A cutoff was further considered to discretize the focal score into values of −2,−1,0,1, and 2, where −2 means complete deletion, −1 means loss, 0 means normal, 1 means gain, 2 means amplification. This dataset contains 185 Basal samples, 549 Luminal A samples, 206 Luminal B samples and 81 HER2+ samples.

### 2.5. Simulation Data

In the simplest hypothetical case, a cancer patient has a single contiguous copy number aberration of some length. As in real cancer patients, this aberration will have copy number values either above or below 0 around the same positive or negative value representing either a gain or a loss. In order to perform well on real data, the method must be capable of distinguishing the copy number aberrations from random noise. To test this, we generate data mimicking patients with single contiguous aberrations and control patients without them (see [14,15]). We used the patients with the aberration as the test set and the rest of the patients as the control set. Specifically, we generated aberrant profiles in the test set with copy number values within the aberration drawn from a normal distribution with mean μ∈{−1,0.6,1} and standard deviation σ∈{0.2,0.22,0.24,…,0.5}. The rest of the copy number values in the aberrant test profiles were drawn from a normal distribution with mean μ=0 and standard deviation σ∈{0.2,0.22,0.24,…,0.5} matching the standard deviation of the aberrant values. The total length of each profile was 20 probes, chosen to match the length of segments in [14]. The length of the contiguous section of aberrant values was λ∈{1,2,3,5,10,15}. Control patient profiles had values drawn from a normal distribution with mean μ=0 and standard deviation σ∈{0.2,0.22,0.24,…,0.5}, matching the standard deviation of their test counterparts. Since there is no guarantee that all patients will contain an aberration, simulations were also run with the MIX parameter which determined the penetrance of the aberration in the population, that is the percentage of patients within the test set that had aberrations. The rest of the patients in the test set consisted of control profiles. The MIX parameter was in {20%,40%,…,100%}. 50 simulations were run for each set of variables and each simulation consisted of 120 patient profiles. 60 of the profiles were in the test set and the other 60 were in the control set. For each combination of parameters, we computed the sensitivity of the TAaCGH method for correctly determining the significance of the test set from the control set. Sensitivity is defined to be $\frac{TP}{TP+FN}$ where TP is the number of true positives and FN is the number of false negatives.

The second set of simulations was performed as well, using the same kind of data as the first set of simulations across all parameters. This time, for each of the 120 patients in a simulation we calculated its persistence curve. Then we calculated the distance from the current patient’s persistence curve to the average curve from the test and control sets. Each was classified as a test profile if its distance from the test persistence curve was smaller than its distance from the control persistence curve and as a control curve otherwise. Given these classifications, sensitivity and specificity was calculated for each simulation. Specificity is defined as $\frac{TN}{TN+FP}$ where TN is true negative and FP is false positive. An average sensitivity and specificity were then computed over the 50 iterations of each simulation with given parameters.

### 2.6. Cancer Subtype Predictive Models

In [15], the authors build a logistic regression predictive model to quantify the predictive power of the detected copy number aberrations. The predictor variables were built using TAaCGH and consisted of two types of predictors. The full chromosome arm copy number changes that were detected by the displacement of the center of mass of the chromosome arm and segment copy number changes whose significance was determined by Betti curves. The center of mass technique was developed to complement the detection of local aberrations by Betti curves in [14]. This approach associated arms 1p, 16p and 16q to the Luminal A subtype, arm 9p to Luminal B, and arms 1p, 2p, 3q, 4p, 5q, 6p, 6q, 8q, 10p, 10q, 12p and 14q to Basal.

We expand this approach to use predictor variables associated with the significant segments detected by lifespan and landscape curves. Next, we compare the predictive power of the models built from each type of curve. In order to make the comparison, we implemented the corresponding logistic models and computed their accuracy. We also computed a confusion matrix for each model.

First, we find the set of chromosome arms *A* with a significant displacement in their centers of mass and the set of maximally non-intersecting significant chromosome sections *K* as described in [14]. Patients are then classified as either positive 1 or negative 0 for indicator variables $I_{a,i}^{CM}$ and $I_{k,i}^{S}$ for significant arms a∈A and sections k∈K. The subscript *i* denotes the patient number. To specifically define these indicator variables, we introduce the notation $P_{i,k}^{Test},P_{i,k}^{Ctrl}$ denoting the average persistence curve from the test and control sets with patient *i* removed from the set of all patients. $P_{i,k}$ denotes the persistence curve of patient *i* on segment *k*. We also introduce the similarity between persistence curves
$$\begin{array}{c} SS_{k,i}^{G}=\sum_{\epsilon }{\left(P_{i,k}-P_{i,k}^{G}\right)}^{2} \end{array}$$
for G∈{Test,Ctrl} and ϵ the filtration parameter. Then the indicator variable $I_{k,i}^{S}$ for patient *i* and section *k* which determines if patient *i* has the aberration for that section is
$$I_{k,i}^{S}=\left\{\begin{array}{cc} 1 & \mathrm{if}\;SS_{k,i}^{Test}<SS_{k,i}^{Ctrl}\\ 0 & \mathrm{if}\;SS_{k,i}^{Test}\geq SS_{k,i}^{Ctrl}. \end{array}\right.$$

Similarly, if the center of mass of the point cloud of the patient is outside the confidence interval for the control group center of mass, then the indicator variable $I_{i,a}^{CM}=1$ otherwise it is 0. Specifically, we have if the center of mass for the arm is a gain for the arm a∈A then
$$I_{a,i}^{CM}=\left\{\begin{array}{cc} 1 & \mathrm{if}\;{\bar{x}}_{i}^{a}>\mu +\frac{t_{\alpha }\sigma }{\sqrt{n}}\;\mathrm{with}\;n-1\;d.f.\\ 0 & \mathrm{otherwise} \end{array}\right.$$
and if the center of mass for the arm is a loss then
$$I_{a,i}^{CM}=\left\{\begin{array}{cc} 1 & \mathrm{if}\;{\bar{x}}_{i}^{a}<\mu +\frac{t_{\alpha }\sigma }{\sqrt{n}}\;\mathrm{with}\;n-1\;d.f.\\ 0 & \mathrm{otherwise} \end{array}\right.$$
where ${\bar{x}}_{i}^{a}=\sum_{probes}\frac{x_{i}^{a}}{n_{a}}$ where $n_{a}$ is the number of probes in the arm *a* and *n* is the number of patients. Next, we fit a logistic regression model for each phenotype and persistent curve type over indicator variables $I_{k,i}^{S}$ and $I_{a,i}^{CM}$ using the patient data from the Horlings dataset and a classification threshold of ≥0.5 for positive classification of the phenotype. Note that a set of $I_{k,i}^{S}$ exists for each type of persistence curve which could lead to some ambiguity, the context of these variables will make it clear which persistence curve they are defined for. The logistic regression model for a specific persistence curve is then
$$\begin{array}{c} \mathrm{Logit}=\mathrm{Intercept}+\sum_{k\in K}w_{k}I_{s,i}^{S}+\sum_{a\in A}w_{a}I_{a,i}^{CM}. \end{array}$$

In order to prevent overfitting, we use forward addition and the Akaike Information Criterion (AIC) for model selection. This criterion introduces a penalty with respect to the number of covariates in the final model
$$AIC:=2k-2\ln\left(\hat{L}\right)$$
where *k* is the number of variables in the model and $\hat{L}$ is the maximum of the likelihood function for the model. Forward addition begins with a null model and adds covariates until the best model is found. We evaluate these models by using leave-one-out cross-validation.

## 3. Bounds on the Distance between Persistence Curves

In this section, we address the statistical properties of persistence curves. Theorem 1 from [20] provides general bounds on the difference between two persistence curves under the $L^{1}$ norm in terms of the bottleneck ($W_{\infty })$ and 1-Wasserstein distances. The stability with respect to these two distances for many persistence curves is summarized in Table 1 from [20]. This table shows that, in general, the Betti curves and lifespan curves are not stable with respect to the bottleneck distance. Applying Theorem 1 from [20] to both these curves does, however, yield bounds on both these curves as computed in [20].

**Theorem** **1**([20])**.**
*Let C and D be persistence diagrams, $W_{\infty }$ denote the bottleneck distance, $n^{C}$ be the number of birth-death pairs in C and $L^{C}$ denote the sum of the lifespans of all birth-death pairs in C. Then*
(1)$$\begin{array}{ccc} ||\beta \left(C,t\right)-{\beta \left(D,t\right)||}_{1} & \leq & 2\max\left(n^{C},n^{D}\right)W_{\infty }\left(C,D\right)+\min\left(L^{C},L^{D}\right) \end{array}$$
(2)$$\begin{array}{ccc} ||\ell \left(C,t\right)-{\ell \left(D,t\right)||}_{1} & \leq & 2\left(L^{C}+L^{D}\right)W_{\infty }\left(C,D\right). \end{array}$$

The main challenge in the application of persistence curves, including Betti and lifespan curves, comes from the fact that small perturbations in the initial point cloud can lead to large changes in the curves [19]. The main result of this section is a bound on the $L^{1}$ norm between two Betti or two lifespan curves built from finite and bounded point clouds with respect to the bottleneck distance.

Consider the bounds on the $L^{1}$ norm between two Betti curves or two lifespan curves from Theorem 1. The bottleneck distance is already stable with respect to small perturbations in the initial point clouds [30], so our bounds will be in terms of it. We need to find bounds on the maximal number of birth-death pairs in a persistence diagram, as well as the maximal lifespan of any birth-death pair.

First, we establish the existence of bounds on these quantities under the given constraints for *i*-dimensional persistent homology. Then we explicitly compute bounds in the case of 1-dimensional persistent homology of the Vietoris–Rips complex. We also compute bounds in the case of 0-dimensional persistent homology for both the Vietoris–Rips and Čech complex. The existence results given here are for Vietoris–Rips and Čech complexes, but essentially the same arguments work for the various forms of witness complexes described in [31]. They also hold for the alpha complex since it is homotopy equivalent to the Čech complex.

**Proposition** **1.**
*Let $P\subseteq R^{n}$ be a finite point cloud, then there exists a bound on the maximal number of birth-death pairs in the persistence diagrams of VR(P,ϵ) and $\overset{ˇ}{C}\left(V,\epsilon \right)$.*


**Proof.** Since *P* is finite, there are a finite number of simplicial complexes that can be built on *P*. Both the VR and Čech filtrations change a finite number of times, hence the persistence diagrams from these complexes have a maximal number of generators. □

Recall we consider reduced 0-dimensional persistent homology to avoid the issue of an infinite lifespan generator.

**Proposition** **2.**
*Let $P\subseteq R^{n}$ be a finite point cloud with diameter d, then the maximal lifespan of a birth-death pair from the i-dimensional persistence diagram of VR(P,ϵ) or $\overset{ˇ}{C}\left(P,\epsilon \right)$ is d.*


**Proof.** Let $P=\{p_{1},\ldots ,p_{k}\}$ be a finite point cloud in $R^{n}$. Consider the epsilon balls B(p,ϵ) for p∈P and ϵ>d. Since the diameter of *P* is *d*, each of these balls must contain all other points in *P*. Therefore, $B\left(p_{i},\epsilon \right)\cap B\left(p_{j},\epsilon \right)\neq \emptyset$ and ${⋂}_{i=1}^{k}B\left(p_{i},\epsilon \right)\neq \emptyset$ so both VR(P,ϵ) and $\overset{ˇ}{C}\left(P,\epsilon \right)$ are (k−1)-simplices. Since simplices are contractible, there is no *i*-dimensional persistent homology for i≥1 and therefore the maximal lifespan is bounded above by *m*. In the 0-dimensional case we consider reduced homology. □

The following theorem can be proved by combining Theorem 1 with Propositions 1 and 2.

**Theorem** **2.**
*Let $P\subseteq R^{n}$ be a finite point cloud with diameter d, then the i-dimensional Betti and lifespan curves of VR(P,ϵ) and $\overset{ˇ}{C}\left(P,\epsilon \right)$ are bounded with respect to small perturbations of P.*


The next results provide explicit bounds on the maximal number of non-homologous generators of 1-dimensional persistent homology in a Vietoris–Rips filtration. To do this, we require that the given point clouds have pairwise distinct distances between points. The following two results were made available to the authors through private correspondence with David Moon [32]. The proofs provided here are different from those provided to the authors.

**Proposition** **3.**
*Let G=(V,E) be a simple graph on n nodes, then the maximal 1st Betti number of the clique complex of G, X(G), is $\left\lfloor \frac{n}{2}\right\rfloor \left\lceil \frac{n}{2}\right\rceil -\left(n-1\right)$.*


**Proof.** The first Betti number of *G* is $\beta_{1}\left(G\right)=\left|E\right|-n+1$. Subtracting the number of triangles *T* in *G* from this quantity yields $\beta_{1}\left(X\left(G\right)\right)=\left|E\right|-n+1-T$. Let $G_{1}$ be a graph containing at least one triangle. Remove an edge from a triangle in $G_{1}$ to obtain $G_{2}$. $G_{2}$ has at least one less triangle than $G_{1}$, but only one less edge so $\beta_{1}\left(X\left(G_{2}\right)\right)\geq \beta_{1}\left(X\left(G_{1}\right)\right)$. The graph *G* for which $\beta_{1}\left(X\left(G\right)\right)$ is maximized must therefore be triangle-free. By Mantel’s theorem [33], the triangle-free graph with the maximal number of edges is the complete bipartite graph $K_{\left\lfloor \frac{n}{2}\right\rfloor ,\left\lceil \frac{n}{2}\right\rceil }$. This graph has $\left\lfloor \frac{n}{2}\right\rfloor \left\lceil \frac{n}{2}\right\rceil$ edges which completes the proof. □

**Proposition** **4.**
*Let $P\subseteq R^{d}$ be a finite point cloud with n vertices such that the pairwise distances between points are distinct. Then the maximal number of birth-death pairs in the 1-dimensional persistence diagram of VR(P,ϵ) is $\left\lfloor \frac{n}{2}\right\rfloor \left\lceil \frac{n}{2}\right\rceil -\left(n-1\right)$.*


**Proof.** The information from a VR filtration can be encoded by a sequence of graphs such that $G_{i}\subseteq G_{i+1}$. Since the distances between points in *P* are pairwise distinct, $G_{i}$ differs from $G_{i+1}$ by a single edge. If adding an edge e=12 to $G_{i}$ to form $G_{i+1}$ completes a triangle 123, then *e* does not birth a new cycle in $H_{1}\left(X\left(G_{i+1}\right)\right)$. To see this note that if *e* completes a cycle say $a_{1},\ldots ,a_{k},1,2$ then this cycle is homologous to $a_{1},\ldots ,a_{k},1,3,2$ in $X(G_{i+1})$ since the two cycles differ by the triangle 123. Since the edges 13 and 32 were in $G_{i}$ the cycle represented by $a_{1},\ldots ,a_{k},1,3,2$ was already in $H_{1}\left(X\left(G_{i}\right)\right)$ and hence *e* did not birth a new cycle. Since triangles do not birth new cycles, any VR filtration which has the maximum number of birth-death pairs in a persistence diagram can have all generators alive at once. Therefore the maximum number of birth-death pairs over the entire filtration is the same as the maximum number of cycles that can be alive at a fixed filtration parameter. Proposition 3 completes the proof. □

Proposition 4 improves a special case of Theorem 3.1 from [34] for n<24, which says that the maximal 1st Betti number of a Vietoris–Rips complex at a fixed filtration value is 5n.

The point clouds considered in this work have a maximum diameter. In particular, if the minimal and maximal copy number ratios are $c_{\min}$ and $c_{\max}$, then all sliding window point clouds with window size *s* are contained in ${\left[c_{\min},c_{\max}\right]}^{s}$. Therefore, the maximum diameter of these point clouds is $d=\sqrt{s}\left(c_{\max}-c_{\min}\right)$.

**Theorem** **3.**
*Let P and $P^{\prime }$ be sliding window point clouds built from copy number ratio data with *window sizes s*. Let P and $P^{\prime }$ each consist of n points. Let C and D be the 1-dimensional persistence diagrams coming from the Vietoris–Rips filtration built on P and $P^{\prime }$. Then*

$$\begin{array}{cc} \left|\right|\beta_{1}\left(C,t\right)-\beta_{1}\left(D,t\right){\left|\right|}_{1} & \leq \left(\left\lfloor \frac{n}{2}\right\rfloor \left\lceil \frac{n}{2}\right\rceil -\left(n-1\right)\right)\left(2W_{\infty }\left(C,D\right)+\sqrt{s}\left(c_{max}-c_{min}\right)\right)\\ \left|\right|\ell_{1}\left(C,t\right)-\ell_{1}\left(D,t\right){\left|\right|}_{1} & \leq 4\left(\left\lfloor \frac{n}{2}\right\rfloor \left\lceil \frac{n}{2}\right\rceil -\left(n-1\right)\right)\sqrt{s}\left(c_{max}-c_{min}\right)W_{\infty }\left(C,D\right). \end{array}$$



In the 0-dimensional case for both the Vietoris–Rips and Čech filtrations, it is clear that the maximal number of connected components in a point cloud with *n* vertices is *n*. This yields the following explicit bounds for $\beta_{0}$ and $\ell_{0}$ curves in the case of copy number variation sliding window point clouds.

**Theorem** **4.**
*Let P and $P^{\prime }$ be sliding window point clouds built from copy number ratio data with *window sizes s*. Let P and $P^{\prime }$ each consist of n points. Let C and D be the 0-dimensional persistence diagrams coming from either Vietoris–Rips or Čech filtrations built on P and $P^{\prime }$. Then*

$$\begin{array}{cc} \left|\right|\beta_{0}\left(C,t\right)-\beta_{0}\left(D,t\right){\left|\right|}_{1} & \leq n\left(2W_{\infty }\left(C,D\right)+\sqrt{s}\left(c_{max}-c_{min}\right)\right)\\ \left|\right|\ell_{0}\left(C,t\right)-\ell_{0}\left(D,t\right){\left|\right|}_{1} & \leq 4n\sqrt{s}\left(c_{max}-c_{min}\right)W_{\infty }\left(C,D\right). \end{array}$$



The bounds from Theorem 4 apply to persistence diagrams that come from a single test patient and a single control patient. In the TAaCGH pipeline the curves from all of the test patients are averaged together, then the curves from all of the control patients are averaged together and finally, these average curves are compared. Since a mean cannot be defined on persistence diagrams, this bound cannot be extended to a bound on average persistence curves.

## 4. Computational Results

### 4.1. Comparison of Performance of Different Persistence Curves on Simulated Data

We used simulations to understand the properties of TAaCGH with various persistence curves as we vary aberrations. As outlined in more detail in Section 2.5 and Figure 3, we created simulated data for patients with single contiguous aberrations defined by the parameters: μ the mean of the aberration, σ the standard deviation of the aberration and λ the length of the aberration. We then studied the sensitivity of the TAaCGH method to distinguish groups of these patients with aberrations from patients with no aberrations. Lastly, we introduced the MIX parameter which determines the percentage of patients in the test set that have the aberration, the rest of the patients have nonaberrant profiles. In particular we consider Betti curves, lifespan curves, and landscape curves within the TAaCGH pipeline on this simulated data, see Figure 4.

We begin by investigating the effect of the MIX parameter on the various topological summaries. In Figure 5, we compare the sensitivity of the lifespan curve to Betti and landscape curves. We do this by fixing the MIX parameter, and considering all simulations with that MIX parameter as the other parameters vary across their ranges as defined in Section 2.5. Then we compute the sensitivity of each persistence curve for each set of simulation parameters. Lastly, we calculate the percentage of all simulations with this fixed MIX parameter for which the sensitivity of the lifespan curve is larger, smaller or equal to the sensitivity of each type of persistence curve.

As the MIX parameter decreases, the lifespan curve outperforms the Betti curve Figure 5a. For the lifespan curve compared to the landscape curves the results are similar Figure 5b–d. As the MIX parameter decreases, for the most part, the percentage of simulations for which the lifespan curve has a higher sensitivity than the landscapes increases or stays the same. In all cases, the lifespan curve has a higher percentage of sensitivity values for which the lifespan sensitivity is higher than the landscape sensitivities. In summary, the lifespan curve outperforms Betti curves and landscape curves as the mix parameter decreases.

Next, we compare the lifespan curve to the Betti curve for a fixed standard deviation σ and vary the other parameters (see Figure 6). For every value of the standard deviation, the lifespan curve has a higher percentage of simulations in which it has a higher sensitivity than Betti and landscape curves. When the lifespan curve is compared to each of the landscape curves in general as the standard deviation increases, the percentage of simulations where the lifespan curve outperforms the landscape curve increases, see Figure 6b,c. The same general trend is visible in Figure 6a, where lifespan curves are compared to Betti curves. The peak percentage is lower than for the comparison to the three landscape curves. Overall, the lifespan curve outperforms the other curve types at higher standard deviations.

The sensitivity values of the lifespan curves compared with the sensitivity values of the Betti and landscape curves with the length λ fixed as the other parameters vary are shown in Figure 7. As expected, the shorter the aberration, the worse each type of curve performs. For example, Figure 7a shows that lifespan curves outperform Betti curves for all aberration lengths. Similarly, Figure 7b–d show that lifespan curves outperform second, third and forth landscape curves respectively except for the third landscape at length 1. It is interesting that the number of cases in which Betti outperforms lifespan is independent from the length of the aberration.

In summary, as we fix the MIX, standard deviation σ or length λ parameters and allow the other parameters to vary, the lifespan curve outperforms the Betti and landscape curves on 0-dimensional persistence. This indicates that lifespan curves are less sensitive to noise in the data of individual patients (as determined by σ and by MIX) than the other persistence curves. Additionally, lifespan curves perform better when aberrations have a small length. The only exception was for Betti curves when the MIX parameter was equal to 1 (Figure 5), where all test patients have the aberration. Therefore, if there is reason to believe that most patients in the test set contain the aberration, then Betti curves could be the better choice.

In the second set of simulations we tested patient classification with Betti curves, lifespan curves and various landscape curves. This is a key step in building subtype classifier models. In Table 1 and Table 2 the average sensitivity and specificity are pictured for simulation data with mean μ=1 and length λ=10 as standard deviation and MIX vary. For both types of curves as the mix parameter increases both the sensitivity and specificity do as well. When comparing the two kinds of curves, lifespan curves have a higher sensitivity than second landscape curves across all parameters. When the mix parameter is at 20% or 40% second landscape curves have better specificity than or equal specificity to lifespan curves. Once the MIX parameter hits 60% lifespan curves have higher specificity.

### 4.2. Comparison of Topological Summaries within the TAaCGH Framework on Horlings Data

As detailed in Section 2.2, JavaPlex, used for the persistent homology computation in [14], differs from the TDA package in *R*. Therefore, we have repeated the study with 0-dimensional Betti curves $\beta_{0}$. Results are shown in Table 3, Table 4, Table 5, Table 6 and Table 7 and indicate that both studies agree for Luminal A, while the new study detected 8p22-p11.1 for Luminal B with Basal patients in the control set, missed 17q21.2-q21.33 from the previously detected segments 17q11.1-q22 for HER2+, while for Basal the new study missed some segments and detected an additional one: 2p23.2-p16.3. The missing segments from the new Betti curve study compared to the old Betti curve study are shown in Table A1. We applied our 3 curves of study, Betti, lifespan and persistence landscape in dimension 0.

No significant regions for the Luminal B subtype were detected in [14] when the Basal subtype was included in the control group. This was hypothesized to be because Luminal B is known to share similar aberrations with other breast cancer subtypes [35]. Therefore, TAaCGH was run with Luminal B as the test set but only HER2+ and Luminal A as the control set in [14]. We repeated this with our three curves of study.

### 4.3. Detecting Breast Cancer Subtypes

In this section, we report regions significant for each breast cancer subtype based on Betti, lifespan, and persistence landscape curves computed with the R TDA software package, GUDHI for computing persistence and Dionysus for detecting generators. Results are compared against TCGA BRCA cohort data set.

#### 4.3.1. HER2

For the HER2+ subtype, the results agree with previous methods. The results are summarized in Table 3. Every method besides lifespan curves missed at least one of the segments detected in the original study. The original study detected 17q11.1-q22, new Betti curves missed 17q21.2-21.33, second landscape functions missed 17q11.1-q22 and third and fourth landscape functions missed 17q21.31-q22.

We used the TCGA BRCA cohort data [29] to validate some of the significant regions that the persistence curves detected. For the HER2 phenotype, all significant sections were contained in chromosome arm 17q. The four chromosome arms with the most aberrant cytobands for the HER2 phenotype are pictured in Figure 8. These are arms 1, 8, 17 and 20.

The most aberrant cytobands in the TCGA BRCA cohort dataset are in arm 17q and are detected by TAaCGH. No cytobands are detected in arms 1,8 or 20 for the HER2 phenotype. One reason for this may be because multiple phenotypes contain significant aberrations in the same chromosome arms in the TCGA BRCA cohort dataset. This can be seen in Figure A1 where the only arm in which the HER2 percentages are significantly higher than the percentages from all patients is in arm 17q.

In our predictive model for the HER2+ subtype, both Betti curves and life-span curves detected 17q12-q21.31, abbreviated 17qs2, with 89% and 92% accuracy, respectively. Landscape 2 and landscape 3, however, detected 17q11.1-q12 abbreviated 17qs1, as the predictor with 90% and 86% accuracy, respectively. These results are consistent since these two regions of the genome overlap in cytoband 17q12, which contains the gene ERBB2. The models for the HER2 subtype are as follows:$$\begin{array}{cc} L_{Betti} & =-2.262+3.766I_{17qs2,i}^{S}\\ L_{lifespan} & =-2.303+20.869I_{17qs2,i}^{S}\\ L_{land2} & =-2.12+20.69I_{17qs2,i}^{S}\\ L_{land3} & =-1.925+3.178I_{17qs1,i}^{S}\\ L_{land4} & =-1.833+20.399I_{17qs1,i}^{S}. \end{array}$$

The confusion matrices of the HER2 models are contained in Table 4.

To further evaluate the HER2 models, we used leave-one-out cross-validation. The average Mean square error (MSE) was 0.104 for the Betti curve model, 0.074 for the lifespan curve model, 0.102 for the second landscape model, 0.130 for the third landscape model and 0.125 for the fourth landscape model. For the HER2 subtype the lifespan curve model has the lowest MSE which agrees with the simulation results.

#### 4.3.2. Luminal A

The significant regions detected by the various methods for the Luminal A subtype are contained in Table 5. There were only two newly detected regions 2q12.1-q21.1 and 5p14.3-p12 which were both detected by lifespan and the third landscape curves. The only other significant region that was detected was 11q22.1-q23.2 which was detected by Betti curves and the third landscape but not lifespan curves or other landscape curves.

The difference between Luminal A compared to all other phenotypes is in Figure A2 in the Appendix A. The Luminal A subtype is associated with 11q22.1-q23.2, 2q12.1-q21.1 and 5p14.3-p12 by the TAaCGH method. These sections were not validated by the TCGA BRCA cohort data at the cytoband level. The arms with the most aberrant cytobands in the TCGA BRCA cohort data along with the methods which detect significant sections within them are pictured in Figure A6. However, there is a clear signal in the Horlings dataset. Figure 9 shows four examples of typical Luminal A patients from 2q12-2q21.1.

There is a large copy number gain between base pairs 125 million to 130 million. Next we build a predictive model using the significant cytobands detected for Luminal A.

In the Luminal A subtype, only Betti, lifespan and landscape 2 curves detected significant copy number changes in segments 2q12.1-q21.1 and 5p14.3-p12 (although 2qs2 was not validated in the TCGA data set). However, only 5p14.3-p12, which we abbreviate to 5ps3, was found to have predictive power. Our results show 80% accuracy. The model from lifespan curve and third landscape curves predicting the Luminal A phenotype are
$$\begin{array}{cc} L_{lifespan} & =-1.743+2.349I_{5ps3,i}^{S}\\ L_{land3} & =-1.587+3.091I_{5ps3,i}^{S}. \end{array}$$

The confusion matrix of this model is contained in Table 6.

The Betti curve logistic regression model did not include any binary predictor variables, so we do not include it.

To evaluate the Luminal A models, we used leave-one-out cross-validation. The average MSE was 0.193 for the lifespan curve model and 0.192 for the third landscape model.

#### 4.3.3. Luminal B

For the case of Luminal B, only one significant section was detected by any of the methods, including the original study. The new Betti 0 study detected 8p22-p11.1. As noted in Section 4.2, it was hypothesized that no significant sections were detected for this subtype because Luminal B is known to contain similar aberrations to the other cancer types. Therefore, we repeated the study while removing the basal patients from the control set. In this case, many new sections were detected, particularly by the lifespan and landscape curves. Cytobands 1q32.1-q41 and 12q21.31-q23.2 were detected by all three methods. The first region, 1q32.1-q41, was a copy number gain in the Horlings dataset. The second region, 12q21.31-q23.2, was driven by one patient in the Luminal B phenotype with significant copy number aberrations, while all other profiles were centered at 0. This matches the TCGA BRCA cohort dataset, where this particular aberration is rare. All newly detected segments are in Table A2 where red indicates newly detected segments.

The difference between Luminal B compared to all other phenotypes is in Figure A3 in the Appendix A. The most aberrant chromosome arms for the Luminal B phenotype in the TCGA BRCA cohort dataset are 1,8,11,17 and 20 and are pictured in Figure A8. The significant regions detected by the TAaCGH method with no basals in the control set are colored in Figure A8. These same arms are pictured in Figure 10 where significant cytobands are colored from the Horlings dataset with Basals in the control set. The new region detected for the Luminal B phenotype is 8p22-8p11.1. This section is validated by the TCGA BRCA cohort data in Figure 11.

The TCGA BRCA cohort data show that a higher percentage of Luminal B patients have a copy number gain of 8p11.21-8p11.23. It also shows that for genes in 8p12-8p22, Luminal B patients have a higher percentage of copy number losses than the other phenotypes. This matches Luminal B patients from the Horlings dataset. Some sample patients from the Horlings dataset are shown in Figure A11 matching the TCGA results.

For the Luminal B subtype, only Betti curves detected a significant segment. This meant there was no comparison to be made between predictive models from different curves, so a predictive model was not built.

#### 4.3.4. Basal

The significant regions detected by the various methods for the Basal subtype are contained in Table 7 with newly detected regions in red. Three new segments are detected as significant compared to the original study: 1q21.1-q25.2, 2p23.2-p16.3 and 23q26.2-q28. The new Betti curve and the lifespan curve both detect 2p23.2-p16.3, but 23q26.2-q28 is only detected by the 4th landscape curve. Both 1q21.1-q25.2 and 23q26.2-q28 are copy number gains in the Horlings dataset, whereas 2p23.2-p16.3 is driven by an undetermined combination of gains and losses within the region. Notably, the second and third landscape functions do not detect any significant segments, suggesting that significance for the Basal subtype in dimension 0 is driven by less persistent connected components.

The significant cytobands for the Basal subtype are in Figure 12 colored by the number of persistence curves that detect them.

The chromosome arms with the most aberrant cytobands for the Basal subtype in the TCGA BRCA cohort dataset are 1,3,5,8,10,12 and are pictured in Figure 13.

Newly detected cytobands 1q21.1-1q24.2, 2p16.3-2p23.2 and 23q26.2-23q28 were detected as copy number gains in the Horlings dataset which is validated by the TCGA BRCA cohort dataset for individual genes within these cytobands. This can be seen in Figure 14, Figure A9 and Figure A10. The difference between basal compared to all other phenotypes are in Figure A4 in the Appendix A.

In Figure A9, we see that over 80% of patients in the TCGA BRCA cohort dataset from the Basal phenotype have copy number gains for most of the genes in cytobands 1q21.1-1q25.2. The other phenotypes only have around 70% of their patients with a copy number gain in these cytobands.

In Figure 14, around 60% of the patients with the Basal phenotype have copy number gains compared to around 10% of patients from the other phenotypes. This supports the detection of 2p16.3-2p23.2 as significant in the Horlings dataset for the Basal phenotype.

In Figure A10, around 30% of the patients with the Basal phenotype have copy number gains compared to around 10% of patients from the other phenotypes. This supports the detection of 23q26.2-23q28 as significant in the Horlings dataset for the Basal phenotype.

In the Basal-like subtype, Betti, lifespan and landscape 4 detected many significant chromosome regions. The variables which had the most predictive power are 1p36.32-p36.11, 1p22.2-p13.3, 3p26.3-p25.1, 4q21.21-q24, 4q31.21-q32.3, 6p25.3-p22.3, 10p15.3-p12.31, 10p12.31-p11.1, 10q23.31-q25.1, 13q21.33-q31.2, 13q31.2-q34 and 15q14-q21.3 abbreviated 1ps1, 1ps10, 3ps1, 4qs4, 4qs10, 6ps1, 10ps1, 10ps3, 10qs6, 13qs6, 13qs8 and 15qs2, respectively, with models:$$\begin{array}{cc} L_{Betti} & =-8.425+2.376I_{1p10,i}^{S}+3.605I_{4qs4,i}^{S}+2.510I_{6ps1,i}^{S}+3.507I_{10ps1,i}^{S}+2.539I_{13qs6,i}^{S}\\ L_{life} & =-4.791+2.289I_{1ps1,i}^{S}+4.183I_{10ps3,i}^{S}+3.711I_{13qs8,i}^{S}+2.689I_{15qs2,i}^{S}\\ L_{land4} & =-4.566+2.681I_{3ps1,i}^{S}+2.331I_{4qs10,i}^{S}+2.803I_{10ps3,i}^{S}+2.528I_{10qs6,i}^{S}. \end{array}$$

The confusion matrices for the Basal subtype models are contained in Table 8.

All three models had an accuracy above 85% (Betti had 92%, lifespan had 86% and landscape 4 had 86%). Consistent with the heterogeneity of the basal subtype the models detected different regions as predictors. All three methods identified chromosome arm 10p as a predictor. Betti detected 10p1 while lifespan and landscape detected 10p3. These are consecutive regions in chromosome 10. Both Betti and Lifespan curves detected regions in chromosome 1 and in chromosome 13. Regions 1qs1 and 1qs10 are far from each other and can be considered as independent predictors. Regions 13qs6 and q8 are consecutive in the genome. Two methods also identified chromosome 4q as a predictor. Betti curves detected region 4qs4 while fourth landscapes detected 4qs10.

To further evaluate the Basal model, we used leave-one-out cross-validation. The average MSE was 0.256 for the Betti curve model, 0.210 for the lifespan curve model, and 0.185 for the fourth landscape model. Since Basal patients share many copy number aberrations with other subtypes of cancer, they tend to be more difficult to distinguish from the other subtypes. This could be the cause for higher MSE values and is a direction for future study.

## 5. Discussion

In this paper, we address the stability of Betti curves and expand the TAaCGH method by incorporating lifespan and persistent landscape curves. We then compare the performance of these curves to identify copy number changes through simulations and apply them to the Horlings dataset. On simulated data, lifespan curves outperform Betti and persistent landscape curves. On the Horlings dataset, all persistence curves are similarly successful at associating chromosome arm segments to phenotypes of breast cancer. Across the four phenotypes, different curves detect different segments, suggesting a complementary approach using the three different methods may provide the most information. From a theoretical perspective, the fact that Betti curves perform comparably to more stable curves in both simulations and on real data is unexpected. It does, however, match results in [23] where Betti curves were shown to be more resistant to certain kinds of noise for the task of image classification than other persistence curves.

The cytoband ranges corresponding to the newly detected segments are pictured in Table 9 organized by type of persistence curve. The ranges are colored by subtype: red for Basal, blue for Luminal A and gray for Luminal B.

All of the newly detected segments were the result of copy number gains except for 2p23.2-p16.3 for the Basal subtype which was inconclusive for being driven by a gain or a loss.

Within the HER2 phenotype, Betti curves detect 17q11.1-q21.31 and 17q21.31-q22, lifespan curves detect 17q11.1-q22 and persistence landscapes all miss one of the four segments in 17q, either 17q11.1-q22 or 17q21.31-q22. Importantly, in each case they detect 17q12 which contains ERBB2, a well-known driver gene for HER2+ breast cancer.

Three chromosome segments were associated with the Luminal A phenotype: 2q12.1-q21.1, 5p14.3-p12 and 11q22.1-q23.2. For this phenotype, persistent landscapes detected all three segments and Betti and lifespan curves detect a subset of these segments. Gains in chromosome arm 11q have been associated with the Luminal A subtype [36], though in different cytobands from the ones detected here. None of the three regions were validated by the TCGA BRCA cohort dataset. This could be due to the fact that only a small percentage of patients (less than 15% with the Luminal A phenotype) have aberrations in any chromosome arm. The highest percentage of aberrations in chromosome arms in the TCGA BRCA cohort dataset occur in arms 1,8,11,17 and 20 which are also the arms with the largest spikes in other phenotypes. In the Horlings dataset, patients in the control set have some type of breast cancer. This means that if multiple phenotypes have a similar signature, TAaCGH may miss these regions.

In the Luminal B phenotype, only one significant segment was detected: 8p22-p11.1. It was detected by Betti curves, but not the other persistence curves. Part of this region was also identified and associated with Luminal B in [37]. In the Horlings dataset most Luminal B patients had a copy number gain of 8p11.21-8p11.23 and a loss of 8p12-8p22. A large number of patients have this same signature in the TCGA BRCA cohort dataset. In general, detecting segments for the Luminal B phenotype is difficult using TAaCGH, because many of its aberrations are shared with the Basal phenotype. To deal with this problem, we removed the Basals from the control group and ran the same experiment. Many significant segments were then detected by the three curves.

For the Basal phenotype, Betti curves detect the most significant segments, followed by lifespan curves and then landscape curves. Only fourth landscape curves detect significant segments. The TCGA BRCA cohort dataset validates many of the detected cytobands. Figure 12 shows the many significant sections identified and also indicates the difficulty of detecting the Basal phenotype since it has many different aberrations which are shared with other phenotypes. In spite of these difficulties, the TAaCGH method identified three new chromosome segments 1q21.1-q25.2, 2p23.2-p16.3 and 23q26.2-q28 associated with the Basal phenotype that are confirmed by the the TCGA BRCA cohort dataset.

The logistic regression predictive models perform fairly similarly across all persistence curves, differing by a few percentage points in accuracy within each phenotype. Even though full chromosome arms were used as potential predictor variables together with chromosome segments, only chromosome segments were used in the final logistic regression models.

Within the HER2 phenotype, the logistic regression equations from Betti, lifespan and second landscape curves all chose the 17q12-q21.31 predictor variable which contains the ERBB2 gene. The third and fourth landscape curves, however, chose 17q11.1-q12. These models are both slightly less accurate than the models that use 17q12-q21.31.

For the Luminal A phenotype lifespan curves and third landscape curves performed with 80% and 83% accuracy on the Horlings dataset. The logistic regression models for each curve used the binary predictor variable associated with 5p14.3-p12.

For the Basal subtype the predictor variables in the models differed significantly from each other. The only repeated predictor variables were 10p15.3-p12.31 and 10p12.31-p11.1. These are adjacent in the genome. In all cases the accuracy is above 85%.

We evaluated the logistic regression models using leave-one-out cross-validation. For the HER2 models and Luminal A models the MSE values were low. In the HER2 case the lifespan curve model had the lowest MSE matching the simulation results. The Basal models had the highest MSE values which could be due to the heterogeneity of the copy number changes in the Basal subtype.

Next we investigated whether the detected cytobands harbor in cancer-related genes. We performed a literature search and also consulted the Sanger Cancer Data Base (COSMIC) [38]. In the Basal subtype, cytoband 1q21.1-q25.2 contains cancer genes HORMAD1, LOC92312, SNG5, TMEM79, CCT3, IQGAP3, HDGF, PRCC in [39], cytoband 2p23.2-p16.3 contains cancer genes PLB1 and WDR43 [40,41] and cytoband 23q26.2-q28 contains the cancer genes ISR4 and FLNA [42,43]. In the case of the Luminal A subtype, cytoband 2q12.1-q21.1 contains gene ECRG4 [44]. Cytoband 5p 14.3-12 contains TERT [45,46,47] and gains in this cytoband have been asociated with this breast cancer [48] and also with recurrence [49]. In the Luminal B subtype, we found cytoband 8p22-p11. These cytobands are commonly associated with the Luminal subtype and contain genes ZNF703, PROSC, BRF2, RAB11FIP1, ASH2L, DDHD2, LETM2 in [39].

## Figures and Tables

**Figure 1 entropy-24-00896-f001:**
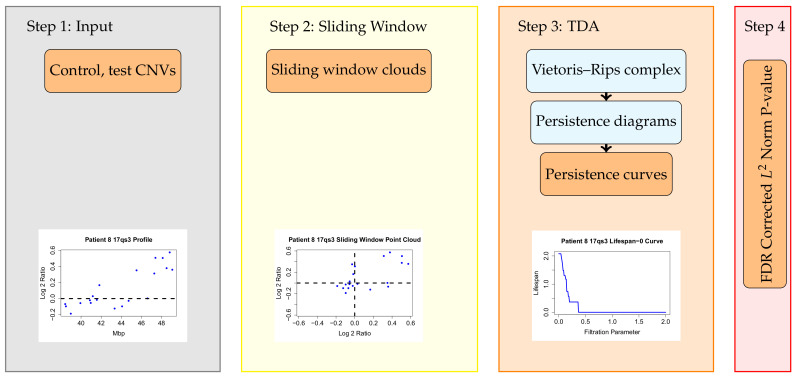
The TAaCGH pipeline. This workflow determines if a segment of the genome is statistically significant for a cancer subtype. Once a particular segment of study is chosen, the TAaCGH pipeline begins. In Step 1 the copy number variation data is separated into control and test patients. Copy number variation data from a single patient is pictured. Next, in Step 2, the data is converted into a sliding window point cloud for each patient. The sliding window point cloud from the sample patient’s data is pictured. In Step 3 a Vietoris–Rips filtration is built on each patient’s point cloud, the persistent homology of each patient’s Vietoris–Rips filtration is computed, recorded in a persistence diagram and summarized into a persistence curve. As an illustration we are using a lifespan curve from the sample patient. Examples of all persistence curves are shown in Figure 2c–e. Lastly, in Step 4, the persistence curves of all patients in the test group and in the control group are averaged and a permutation test is run on the $L^{2}$ norm of these averaged persistence curves.

**Figure 3 entropy-24-00896-f003:**
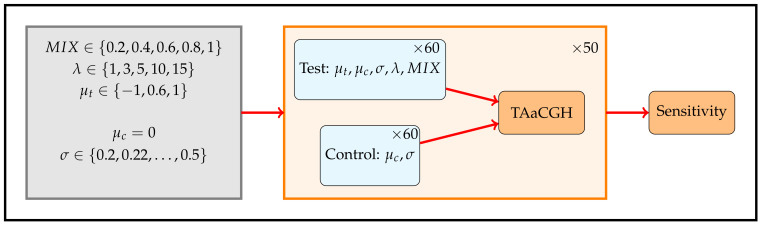
Simulation design. Simulations are designed to test the ability of TAaCGH with various persistence curves to distinguish a set of patients with a single contiguous aberration from a set of patients without them. Each patient has 20 probes. Test patients with aberrations of length λ have copy number values sampled from a normal distribution with mean $\mu_{t}$ and standard deviation σ. The remaining copy number values for test patients are sampled from a normal distribution with mean $\mu_{c}=0$ and standard deviation σ. Control patients have all copy number values sampled from a normal distribution with mean $\mu_{c}=0$ and standard deviation σ. The MIX parameter controls the percentage of patients in the test set that have aberrations, the remaining patients in the test set have data drawn from a normal distribution with mean $\mu_{c}=0$ and standard deviation σ. For each set of parameters, we ran 50 simulations. Each simulation consisted of a total of 120 patients, 60 in the test set and 60 in the control set. The parameters varied over the following values $\mu_{t}\in \left\{-1,0.6,1\right\},$σ∈{0.2,0.22,⋯,0.5},λ∈{1,3,5,10,15} and MIX∈{20%,40%,60%,80%,100%}.

**Figure 4 entropy-24-00896-f004:**
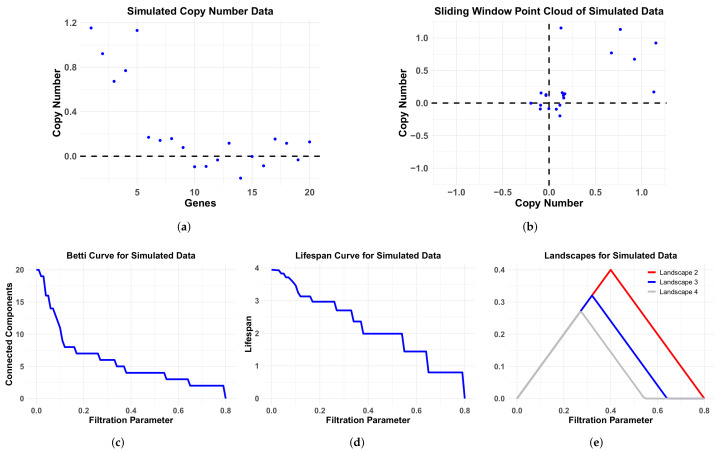
Simulation data and associated persistence curves. Simulated CNA data (**a**), corresponding sliding window point cloud (**b**), Betti curve (**c**), lifespan curve (**d**) and landscape curves (**e**) for a hypothetical cancer patient on a segment of 20 probes with a single length λ=5 contiguous aberration with aberration mean μ=1 and standard deviation σ=0.2. The nonaberrant probes are sampled from a distribution with mean μ=0 and standard deviation σ=0.2.

**Figure 5 entropy-24-00896-f005:**
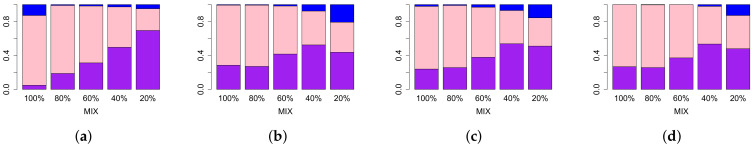
Comparison of sensitivity values of different approaches in TAaCGH with respect to varying MIX parameter on persistence of simulated data in dimension zero. Figure shows comparisons between lifespan $\ell_{0}$ curve and Betti $\beta_{0}$ (**a**), second landscape $\lambda_{2}$ (**b**), third landscape $\lambda_{3}$ (**c**), fourth landscape $\lambda_{4}$ curves (**d**). For a fixed value of the MIX parameter, we compute the sensitivity of our method with one of the persistence curves for each set of simulations as the standard deviation σ, the mean μ and the length λ vary over all possible values detailed in Section 2.5. The height of each bar represents the percentage of those simulations where the sensitivity of the lifespan curve was bigger (purple), equal to (pink) and less than the sensitivity (blue) of other persistence curves.

**Figure 6 entropy-24-00896-f006:**
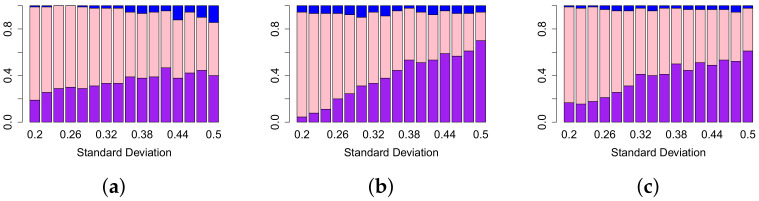
Comparison of sensitivity values of different approaches in TAaCGH with respect to varying the standard deviation σ on persistence of simulated data in dimension zero. The figure shows comparisons between lifespan $\ell_{0}$ curve and Betti $\beta_{0}$ (**a**), second landscape $\lambda_{2}$ (**b**), fourth landscape $\lambda_{4}$ curves (**c**). For a fixed value of the standard deviation σ, we compute the sensitivity of our method with one of the persistence curves for each set of simulations as the mean μ, length λ and MIX parameters vary over all possible values detailed in Section 2.5. The height of each bar represents the percentage of those simulations where the sensitivity of the lifespan curve was bigger (purple), equal to (pink) and less than the sensitivity (blue) of the other persistence curve. Note that the third landscape $\lambda_{3}$ behaves similarly to the second landscape.

**Figure 7 entropy-24-00896-f007:**
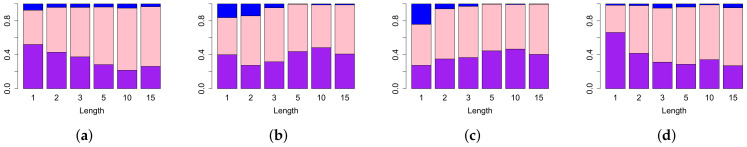
Comparison of sensitivities of different approaches in TAaCGH with respect to varying the length λ on persistence of simulated data in dimension zero. The figure illustrates differences between lifespan $\ell_{0}$ curve and Betti $\beta_{0}$ (**a**), second landscape $\lambda_{2}$ (**b**), third landscape $\lambda_{3}$ (**c**), fourth landscape $\lambda_{4}$ curves (**d**). For a fixed value of the MIX parameter, we compute the sensitivity of our method with one of the persistence curves for each set of simulations as the standard deviation σ, the mean μ and the mix parameter varies over all possible values detailed in Section 2.5. The height of each bar represents the percentage of those simulations where the sensitivity of the lifespan curve was bigger (purple), equal to (pink) and less than the sensitivity (blue) of other persistence curves.

**Figure 8 entropy-24-00896-f008:**
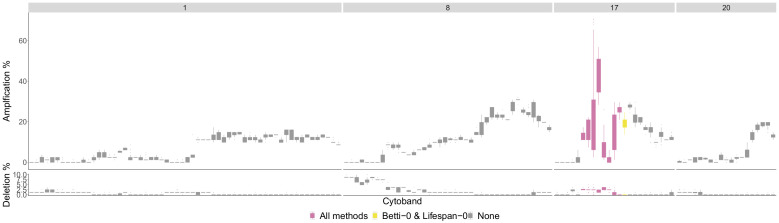
HER2 phenotype most aberrant cytobands in TCGA BRCA cohort Data Cytobands with gains and losses in the TCGA BRCA cohort dataset [29] for the HER2 phenotype. The chromosome arms 1,8,17 and 20 are included since they had above 10% of patients with aberrations in genes in those cytobands on average. The colors indicate which persistence curves detected those cytobands as significant in [22]. Each gene within a cytoband has a score of −2,−1,0,1,2 in the TCGA BRCA cohort dataset indicating major deletions, mild deletions, normal, mild copy number gain and major copy number gain. In the top plot for each cytoband we plot a boxplot of the percentages of HER2 patients with a score of 2 for each gene in the cytoband. The bottom plot is the same except we calculate the percentage of genes with a score −2.

**Figure 9 entropy-24-00896-f009:**
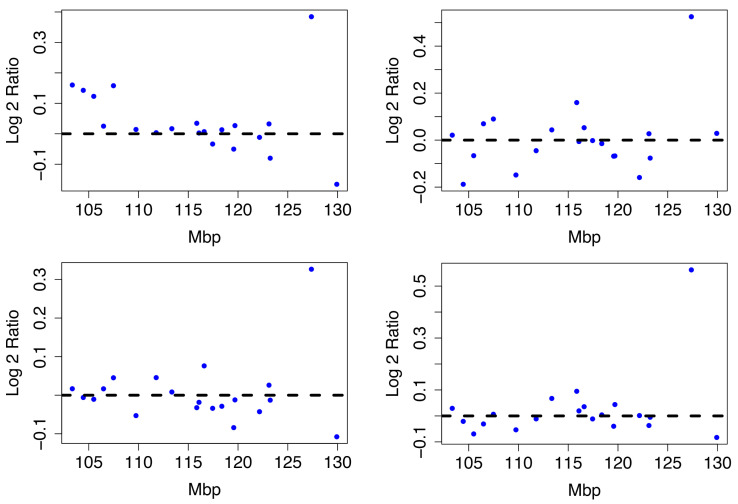
Luminal A Patient Profiles. Four Luminal A patient profiles from the Horlings dataset on cytobands 2q12-2q21.1. All share a significant copy number gain between 125 and 135 Mbp.

**Figure 10 entropy-24-00896-f010:**
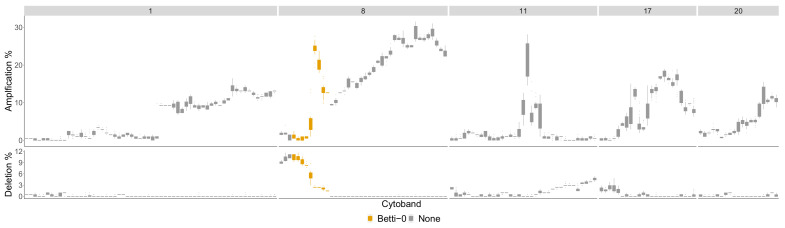
Luminal B most aberrant cytobands in TCGA BRCA cohort Data. The chromosome arms 1,8,11,17 and 20 are included above 10% of patients had aberrations in the genes within this cytoband on average. The colors indicate which persistence curves detected those cytobands as significant in [22] for the Luminal B phenotype with no Basals in the control group. Each gene within a cytoband has a score of −2,−1,0,1,2 in the TCGA BRCA cohort dataset [29] indicating major deletions, mild deletions, normal, mild copy number gain and major copy number gain. In the top plot for each cytoband we plot a boxplot of the percentages of Luminal B patients with a score of 2 for each gene in the cytoband. The bottom plot is the same except we calculate the percentage of genes with score −2.

**Figure 11 entropy-24-00896-f011:**
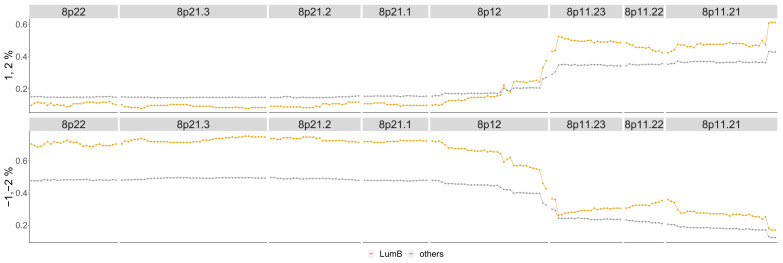
Luminal B phenotype cytobands 8p22-8p11.21. Top graph: Shows the percentage of patients in the Luminal B phenotype with either a 1 or 2 score from the TCGA data (orange) as well as the percentage of all other phenotypes with these scores (gray). Bottom graph: Shows the same as the top graph but for scores of −1, −2.

**Figure 12 entropy-24-00896-f012:**
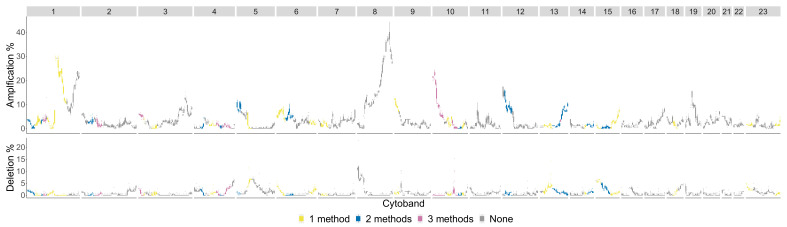
Basal phenotype cytobands in TCGA BRCA cohort dataset. The colors indicate how many persistence curve methods detected that particular cytoband in the Horlings dataset [22]. Each gene within a cytoband has a score of −2,−1,0,1,2 in the TCGA BRCA cohort dataset [29] indicating major deletions, mild deletions, normal, mild copy number gain and major copy number gain. In the top plot for each cytoband we plot a boxplot of the percentages of Basal patients with a score of 2 for each gene in the cytoband. The bottom plot is the same except we calculate the percentage of genes with a score −2.

**Figure 13 entropy-24-00896-f013:**
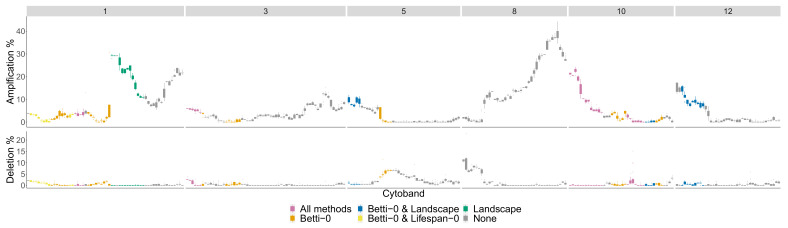
Basal phenotype most aberrant cytobands in TCGA BRCA cohort dataset. The chromosome arms 1,3,5,8,10, and 12 are included since above 10% of patients had aberrations in the genes in this cytoband. The colors indicate which persistence curves detected those cytobands as significant in [22]. Each gene within a cytoband has a score of −2,−1,0,1,2 in the TCGA BRCA cohort dataset [29] indicating major deletions, mild deletions, normal, mild copy number gain and major copy number gain. In the top plot for each cytoband we plot a boxplot of the percentages of Basal patients with a score of 2 for each gene in the cytoband. The bottom plot is the same except we calculate the percentage of genes with a score −2.

**Figure 14 entropy-24-00896-f014:**
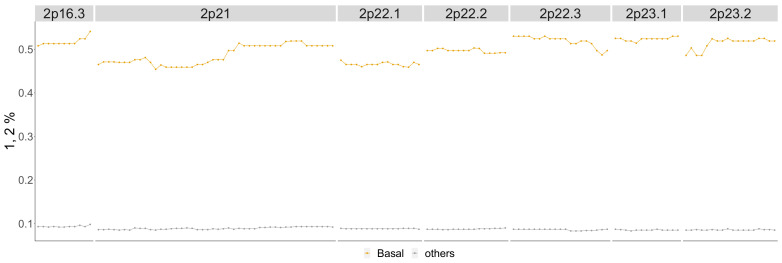
Basal phenotype cytobands 2p16.3-2p23.2. Shows the percentage of patients in the Basal phenotype with either a 1 or 2 score from the TCGA BRCA cohort dataset (orange) as well as the percentage of all other phenotypes with these scores (gray).

**Table 1 entropy-24-00896-t001:** Average sensitivity and specificity of lifespan curves for patient classification. The length of aberration in aberrant profiles, λ, is fixed at 10 of 20 total probes for this table.

Lifespan Curves										
μ=1	20% mix		40% mix		60% mix		80% mix		100% mix	
σ=0.2	32.00%	85.00%	42.00%	96.00%	60.00%	99.00%	80.00%	100.00%	99.00%	100.00%
σ=0.5	49.00%	62.00%	56.00%	70.00%	63.00%	76.00%	71.00%	79.00%	78.00%	84.00%
Total	40.50%	73.50%	49.00%	83.00%	61.50%	87.50%	75.50%	89.50%	88.50%	92.00%
	TPR	SPC	TPR	SPC	TPR	SPC	TPR	SPC	TPR	SPC

**Table 2 entropy-24-00896-t002:** Average sensitivity and specificity of second landscape curves for patient classification. The length of aberration in aberrant profiles, λ, is fixed at 10 of 20 total probes for this table.

Landscape 2										
μ=1	20% mix		40% mix		60% mix		80% mix		100% mix	
σ=0.2	27.00%	92.00%	41.00%	98.00%	59.00%	99.00%	78.00%	100.00%	93.00%	100.00%
σ=0.5	49.00%	60.00%	48.00%	68.00%	53.00%	70.00%	58.00%	73.00%	63.00%	76.00%
Total	38.00%	76.00%	44.50%	83.00%	56.00%	84.50%	68.00%	86.50%	78.00%	88.00%
	TPR	SPC	TPR	SPC	TPR	SPC	TPR	SPC	TPR	SPC

**Table 3 entropy-24-00896-t003:** HER2 Phenotype: Cytobands detected by 0-dimensional Betti, lifespan and persistence landscape curves for the HER2 subtype on the Horlings dataset [22] using the *R* TDA package.

	HER2+
Betti-0	17q 11.1-q21.31, 17q21.31-q22
Lifespan-0	17q11.1-q22
Landscape	$\lambda_{2}$: 17q11.1-q21.31, $\lambda_{3}$: 17q11.1-q21.33, $\lambda_{4}$: 17q11.1-q21.33

**Table 4 entropy-24-00896-t004:** HER2 Logistic Regression: The confusion matrices and accuracy of logistic regression models built to predict the HER2 phenotype from Betti, lifespan and landscape curves.

Betti HER2		Lifespan HER2		Landscape 2 HER2		Landscape 3 HER2		Landscape 4 HER2	
9	5	9	5	8	6	7	7	6	8
2	48	0	50	0	50	2	48	0	50
Accuracy: 89%		Accuracy: 92%		Accuracy: 90%		Accuracy: 86%		Accuracy: 88%	

**Table 5 entropy-24-00896-t005:** Luminal A Phenotype: Cytobands detected by 0-dimensional Betti, lifespan and persistence landscape curves for the Luminal A subtype on the Horlings dataset [22] using the *R* TDA package.

	Luminal A
Betti-0	11q 22.1-q23.2
Lifespan-0	2q12.1-q21.1, 5p14.3-p12
Landscape	$\lambda_{3}$: 2q12.1-q21.1, 5p14.3-p12, 11q22.1-q23.2

**Table 6 entropy-24-00896-t006:** Luminal A Logistic Regression: The confusion matrices and accuracy of the logistic regression models built to predict the Luminal A phenotype from lifespan curves and third landscape curves.

Luminal A Lifespan		Luminal A Landscape 3	
11	7	9	9
6	40	2	44
Accuracy: 80%		Accuracy: 83%	

**Table 7 entropy-24-00896-t007:** Basal phenotype: Cytobands detected by 0-dimensional Betti, lifespan and persistence landscape curves for the Basal subtype on the Horlings dataset [22] using the *R* TDA package.

	Basal
Betti-0	1p 36.32-p33, 1p32.3-p31.1, 1p22.2-p12, 2p23.2-p16.3, 2p15-p11.2, 3p26.3-p24.3, 3p21.2-p13, 4p15.1-p11, 4q21.21-q34.1 5p15.33-p15.1, 5q11.1-q13.1, 6p25.3-p22.1, 6p21.33-p11.2, 6q24.1-q27, 7p21.3-p14.2, 9p24.3-p22.3, 10p15.3-p11.1, 10q21.1-q22.1, 10q22.2-q26.11, 12p13.31-p11.21, 13q12.2-q31.2, 13q31.2-q34, 14q24.3-q32.33, 15q11.2-q22.31, 15q23-q26.3, 18q12.1-q21.2, 23p22.33-p11.21
Lifespan-0	1p36.32-p36.11, 1p32.3-p31.1, 2p23.2-p16.3, 2p15-p11.2, 3p26.3-p25.1, 4q24-q27, 4q28.3-q31.3, 4q31.3-q34.1, 6p21.33-p11.2, 10p15.3-p11.1, 10q23.1-q24.2, 13q21.1-q31.2, 15q14-q22.31, 23p13.2-p12
Landscape	$\lambda_{4}:$ 1p32.1-p31.1, 1q21.1-q25.2, 2p15-p11.2, 3p26.3-p25.1, 4p15.1-p11, 4q24-q28.3, 4q31.21-q34.1, 5p15.33-p15.1, 10p15.3-p12.31, 10p12.31-p11.1, 10q23.1-q25.1, 12p13.31-p11.21, 13q31.2-q34, 14q31.3-q32.33, 23p22.33-p21.3, 23q26.2-q28

**Table 8 entropy-24-00896-t008:** Basal Logistic Regression: The confusion matrices and accuracy of logistic regression models built to predict the Basal phenotype from Betti, lifespan and landscape curves.

Betti Basal		Lifespan Basal		$\lambda_{4}$ Basal	
17	2	14	5	16	3
3	42	4	1	6	39
Accuracy: 92%		Accuracy: 86%		Accuracy: 86%	

**Table 9 entropy-24-00896-t009:** The cytobands corresponding to the new regions detected by Betti curves, lifespan curves, and landscape curves compared to the regions detected in [14]. Red indicates new cytobands associated with the Basal subtype, blue with Luminal A and gray Luminal B.

	Cytoband Ranges of Newly Detected Segments
Betti-0	2p23.2-p16.3, 8p22-p11.1
Lifespan-0	2p23.2-p16.3, 2q12.1-q21.1, 5p14.3-p12
Landscape	$\lambda_{3}$: 2q12.1-q21.1, 5p14.3-p12, $\lambda_{4}$: 1q21.1-q25.2, 23q26.2-q28

## Data Availability

The Horlings data and scripts used for this project can be found at https://github.com/Jkaslam/TDA-Cancer-Genomics, accessed on 15 June 2022.

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
