# Peer review of "TAaCGH Suite for Detecting Cancer—Specific Copy Number Changes Using Topological Signatures"

_entropy, 2022, doi:10.3390/e24070896_

Round 1

Reviewer 1 Report

In this article, the authors propose to study copy number changes through the lens of topological data analysis (TDA). The idea is that copy number variations associated to consecutive segments of the genome can be efficiently characterized by first embedding them as a point cloud in the Euclidean plane with sliding window embedding, and then computing the Vietoris-Rips persistent homology of this point cloud. Then, one can vectorize the corresponding persistence diagram using persistence curves, and finally derive statistical tests based on the L2 norms of these curves to identify significant segments associated to various phenotypes, such as breast cancer subtypes. In this work, the authors use and compare a variety of curves, namely the Betti, landscapes and (new) lifespan curves. They show that these curves enjoy stability properties for finite point clouds, and that they are complementary, in the sense that each curve can identify its specific segments.

Overall, I am quite positive about this work. I think the writing is good and easy to understand, and that the results are quite interesting (even though I am not an expert in the study of copy number variations). The experiments look good, with a nice simulation experiment motivating the use of lifespan curves, and applications to real-world data where known segments were recovered, and new segments were identified.

I just have some comments that I would like the authors to discuss prior to publication:

---It looks like Theorem 3 could be easily generalized to sliding windows of any size w, by replacing sqrt(2) with sqrt(w). 

---The stability result for Betti curves is not an usual TDA stability result, since there is a constant in the upper bound, so the Betti curves can still be at positive distance from each other, even if the persistence diagrams are the same. This should probably be mentioned in the text. 

---Overall, it feels like the stability results are more about expliciting (already known) upper bounds when the point clouds are finite and the filtration is Vietoris-Rips, rather than brand new stability properties. This should be made more clear in the text.

---Some more details could be provided for the simulation experiment, especially concerning the sensitivity of the test. For instance, what test statistic is used? I understand how to get a p-value using random permutations, but I do not fully understand what statistic is computed (based on the persistence curves) for computing the sensitivity.

---The comparison between curves is well done and interesting. It would be great to also add the persistence silhouette (https://arxiv.org/abs/1312.0308) in the comparison, since it is another common persistence curve.

---TDA practitioners are now starting to automatically choose the best filtration / vectorization method with respect to the data (http://proceedings.mlr.press/v119/hofer20b.html, https://proceedings.mlr.press/v108/carriere20a.html, http://proceedings.mlr.press/v139/carriere21a/carriere21a.pdf), in order to avoid choosing between several filtrations or vectorizations. Thus, I wonder whether it would be possible to define an optimization loss that would identify the best vectorization among several persistence curves. For instance, one could use, for each segment, a loss based on the L2 norm between the control and test groups, in order to identify the persistence curve that best separates the two.

Author Response

We would like to thank the referee for the suggestions and comments (all
have been addressed), see attachment.

Reviewer 2 Report

Aslam et al. present an extension of their topology-based method TAaCGH for identifying recurrent copy number alterations in large cancer studies, which was first introduced in DeWoskin et al. 2011. Specifically, they expand TAaCGH by introducing other summary statistics, including lifespan and persistent landscape curves, in addition to Betti curves. They test these variants of the method on two large datasets of breast cancer (the Horlings and TCGA cohorts), as well as on simulated data, and find several new segments with recurrent copy number alterations in this cancer. In addition, the authors derive some bounds for these summary statistics. Overall, the article is interesting and presents a substantial amount of new material which will be useful for researchers interested in the application of topological data analysis to cancer genomics. In my opinion, the article is suitable for publication in this special issue. There are however several aspects that the authors might consider for improving the manuscript:

  • The sensitivity of TAaCGH is estimated in section 4 using simulated data. However, the sensitivity of a test is only meaningful in the context of its specificity (e.g. rejecting always the null model would lead to a sensitivity of 100%). It would be very helpful if the authors also compute the specificity of the TAaCGH in these simulations as a function of the parameters.
  • The authors build predictive models using logistic regression. However, the same set that is used to fit the model is also used to compute the accuracy of the predictions. Implementing a cross-validation scheme or training the classifier using the Horling dataset and estimating the accuracy in the TCGA dataset would lead to more robust estimates of the accuracy and would be informative about the generalizability of the model.
  • TAaCGH uses the L2 norm in its test statistics. However, the bounds derived in section 3 are based on the L1 norm. Although these bounds cannot be applied to average persistence curves, a more extended discussion on how the bounds relate to TAaCGH would improve the cohesiveness of the presentation.
  • One of the main results of this paper is the discovery of new recurrent copy number alterations (Table 7). Does any of these cytobands contain known oncogenes (e.g. from existing databases)? This information would add further biological context to their results.

There are also several typos:

  • Line 29: the acronym aCGH needs to be introduced.
  • Line 102: “data from the chromosome arm of each into” -> “data from each chromosome arm into”
  • Line 109: “is pictured” -> “are pictured”
  • Line 129: “copy number values estimated” -> “copy number values were estimated”
  • Line 179: “Betti and or two” -> “Betti or two”
  • Table 2: the entries of the confusion matrix need to be defined (e.g. TP, FP, FN, TN)
  • The “qs” abbreviation is first used on line 326, but it is not introduced there.

Author Response

(The authors gave the same response as above.)
